# Benchmarking the Attribution Quality of Vision Models

**Robin Hesse**[1] **Simone Schaub-Meyer**[1,2] **Stefan Roth**[1,2]

[1]Department of Computer Science, Technical University of Darmstadt  [2]hessian.AI
{robin.hesse, simone.schaub, stefan.roth}@visinf.tu-darmstadt.de

## Abstract

Attribution maps are one of the most established tools to explain the functioning of computer vision models. They assign importance scores to input features, indicating how relevant each feature is for the prediction of a deep neural network. While much research has gone into proposing new attribution methods, their proper evaluation remains a difficult challenge. In this work, we propose a novel evaluation protocol that overcomes two fundamental limitations of the widely used incremental-deletion protocol, *i.e.*, the out-of-domain issue and lacking inter-model comparisons. This allows us to evaluate 23 attribution methods and how different design choices of popular vision backbones affect their attribution quality. We find that intrinsically explainable models outperform standard models and that raw attribution values exhibit a higher attribution quality than what is known from previous work. Further, we show consistent changes in the attribution quality when varying the network design, indicating that some standard design choices promote attribution quality.[1]

## 1 Introduction

In recent years, deep neural networks (DNNs) have become an integral part of computer vision. However, their strong performance goes hand in hand with the development of increasingly complex models, surpassing the boundaries of human understanding. Consequently, DNNs are generally less trustable than traditional methods of computer vision, rendering their application in safety-critical domains difficult. To address such issues, methods of explainable artificial intelligence (XAI) aim to unravel the inner workings of neural architectures. In the context of computer vision, *attribution maps* have proven to be particularly important explanation types [60]. They assign an importance score to each input pixel of a DNN to visualize what parts of the input image have been most relevant for the final prediction of a classification model. While various attribution methods have been proposed, their evaluation is challenging and an active research area. As a consequence, interesting questions, such as how different design choices of DNNs affect their attribution quality, have thus been underexplored.

One of the most important protocols for evaluating attribution maps is the incremental-deletion protocol [5, 54]. Here, pixels or patches of an input image are incrementally deleted in ascending or descending order of their corresponding attribution values to measure the effect on the model output for a specific class. Although widely used in many different flavors, these protocols come with the following two fundamental limitations: First, deleting pixels or patches introduces domain changes that can interfere with the evaluation metrics, as output changes can be due to the removal of important information or due to the change of domain [29]. Second, as model outputs are highly influenced by the properties of the used model, *e.g.*, its calibration, these protocols do not allow for inter-model comparisons. As a result, one cannot easily measure the effect of different model design choices on

---

[1]Code available at github.com/visinf/idsds

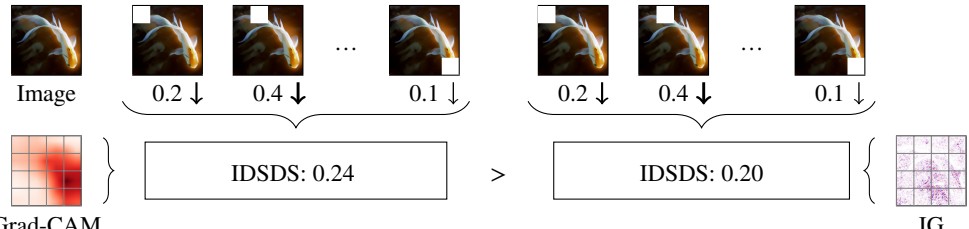

Figure 1: *Illustration of our in-domain single-deletion score (IDSDS) for evaluating the correctness of attribution maps.* We obtain "ground-truth" importance scores for each non-overlapping image patch by feeding images with deleted patches (shown in white) through the model under inspection and measuring the output drop of the model's logit for the target class. The larger the drop (denoted by numbers and arrow width), the more important is the patch for the model. Next, we divide the attribution map into the corresponding patches and measure the attribution sum per patch. Finally, we obtain our IDSDS by computing the Spearman rank-order correlation between the output drops and the corresponding patch-attribution sums. To ensure that all image interventions are in-domain, we fine-tune the model under inspection on images with deleted patches before the evaluation.

the attribution quality, which becomes especially important with the accelerating development of intrinsically explainable vision models [7–9, 27, 48].

With this work, we overcome these two limitations, allowing us to evaluate the effect of different design choices of popular image classification models, which are widely used as backbones, on their attribution quality without suffering from out-of-domain issues. Specifically, *(i)* we propose a protocol based on our in-domain single-deletion score (IDSDS), see Figure 1, that allows for *inter-model* comparisons and an *exact* alignment of the train and test domains *without* the issue of class information leakage. Using our proposed protocol, we provide a novel analysis *(ii)* by evaluating 23 attribution methods on ImageNet models and *(iii)* by systematically studying how different design choices of popular backbone models affect their attribution quality. Our analysis highlights that intrinsically explainable models produce significantly more correct attributions than standard models and that raw attribution values often outperform absolute attributions, which contradicts findings from previous work [66]. Further, we observe clear changes in the attribution quality when varying the network design, indicating that some design choices have a consistently positive effect on the attribution quality. Finally, to the best of our knowledge, we are the first to empirically confirm the often-mentioned accuracy-explainability trade-off in a large-scale study.

## 2 Related work

**Attribution methods.** Given a DNN and an input, attribution methods assign each input feature an importance score, indicating how relevant that feature is for the final prediction of the DNN. *Perturbation-based attribution methods* achieve this goal by perturbing inputs or neurons and measuring the output change of the model [36, 47, 49, 68, 71, 72]. *Backpropagation-based attribution methods* backpropagate an importance signal, *e.g.*, the gradient, to the input to measure the relevance of each feature [1, 4, 5, 14, 19, 38, 58, 60, 64]. While above attribution methods have been developed for the post-hoc attribution of standard models, there are also *intrinsically explainable models* specifically designed to produce high-quality attribution maps [7–9, 27, 48].

**The correctness of attribution maps.** One particularly important metric to evaluate existing attribution methods is *correctness*, *i.e.*, how faithfully an attribution map reflects the model behavior [41].

The **incremental-deletion protocol** follows the idea that intervening on pixels with larger attribution scores should have a stronger effect on the model output than intervening on pixels with lower attribution scores [5, 12, 21, 24, 27, 29, 39, 47, 50, 54, 58, 62]. One of its initial instantiations is the pixel flipping protocol [5], where pixels in MNIST [34] images are incrementally flipped based on their attribution order to measure the impact on the model under inspection. Samek et al. [54] extend [5] by considering different sets of locations and different interventions, *e.g.*, local blurring. Following this protocol, a multitude of different instantiations has been used with slight variations, such as different baselines, patch sizes, and orderings [12, 21, 27, 47, 50, 58, 62].

Table 1: *Overview of existing evaluation protocols.* Our IDSDS protocol is the first to achieve an exact alignment of the train and test domains without suffering from information leakage while allowing for inter-model comparisons and working on natural data.

| Protocol | Domain alignment | No inf. leakage | Inter-model comp. | Natural data |
|---|---|---|---|---|
| Incremental deletion [54] | ✗ | N/A | ✗ | ✓ |
| Single deletion [55] | ✗ | N/A | ✓ | ✓ |
| ROAR [29] | ✓ | ✗ | ✗ | ✓ |
| FunnyBirds [28] | ✓ | ✓ | ✓ | ✗ |
| **IDSDS (ours)** | ✓ | ✓ | ✓ | ✓ |

A limitation of these protocols is the out-of-domain (OOD) issue occurring when perturbing images [12, 24, 29]. As a consequence, it remains unclear if the observed output changes are due to important features being removed or due to the resulting domain changes. To minimize this issue, generative models can be used for infilling the deleted areas [12]. Hooker et al. [29] proposed ROAR, where the model is re-trained after each degradation level of the incremental-deletion protocol to align the train and test domains. However, this leads to different models for each attribution method, impeding a fair comparison. Further, the masking of pixels can leak class information, which can also interfere with scores [51]. Briefly, masking pixels based on image content can introduce new shape information in the image, and thus, the "removal" of information actually adds new information. Additionally, as the incremental-deletion protocol is directly dependent on the model output, and thus, a mere change in the model calibration can affect the score, it is also not well suited for comparing attribution methods on *different* models. Böhle et al. [10] address this issue by only evaluating the 250 most confidently and correctly classified images. However, this introduces a selection bias that could alter the scores as these images might not be representative for *all* images. Contrary to existing work, our in-domain single-deletion score proposed in Section 3 achieves an *exact* alignment of the train and test domains *without* suffering from information leakage and *without* requiring different models for evaluating multiple attribution methods. Additionally, it is well suited for comparing *different* models regarding their attribution correctness.

Another established tool to measure the correctness of attribution methods is the **single-deletion protocol**, where individual features or feature groups are deleted to measure the correlation or error between the resulting output changes and the corresponding attribution scores [3, 16, 23, 28, 44, 55, 56, 70]. For example, Selvaraju et al. [55] measure the rank correlation between image occlusions [68], *i.e.*, the probability drops occurring when masking the image with a sliding window, and the corresponding attribution scores. Similarly, Alvarez-Melis and Jaakkola [3] measure the correlation between the output drops from deleted image regions and the corresponding attribution scores.

While existing single-deletion protocols are similar to our approach, none of the previous variations that work on natural images achieve an *exact* alignment of the train and test domains as we do, which, however, is critical (*cf*. Figure 2 (c) and [12, 24, 28, 29]). Further, we utilize our protocol to provide a novel analysis of how the model design affects the attribution quality of the final model.

Somewhat orthogonal to the above approaches, the **controlled synthetic data check protocol** [2, 15, 28, 32, 41, 43, 52] uses synthetic datasets to evaluate if the explanation of a model aligns with the dataset features that are known to be important by design. For example, in the an8Flower dataset [43], the class-specific features are known, and in the FunnyBirds dataset [28], individual bird parts are deleted to find the subset of important parts.

To conclude, established evaluation protocols on real images suffer either from misaligned train and test domains or from information leakage. Further, incremental-deletion protocols cannot be used for inter-model comparisons, and single-deletion protocols on real images have so far not been used for inter-model comparisons. Table 1 gives an overview of the most important protocols.

## 3 In-Domain Single-Deletion Score (IDSDS)

Motivated by the two fundamental limitations of deletion-based protocols for assessing attribution correctness on natural images, *i.e.*, OOD issues, respectively information leakage, and/or no inter-model comparison, we propose an evaluation scheme that addresses these problems (*cf*. Figure 1).

**IDSDS.** Given a DNN $f$ and a dataset of $N$ images $\{x_n \in \mathbb{R}^{C \times H \times W} \mid n = 1, \dots, N\}$ of target classes $\{t_n \mid n = 1, \dots, N\}$, an attribution map $\mathcal{A}(f_{t_n}, x_n) \in \mathbb{R}^{H \times W}$ displays the contribution of each pixel in $x_n$ to the target logit output $f_{t_n}(x_n)$. In our IDSDS protocol for evaluating the correctness of attribution methods, we divide each input image $x_n$ into $P$ non-overlapping patches $\{p_m \mid m = 1, \dots, P\}$ and measure the Spearman rank-order correlation between the attribution sum in each patch $p_m$ and the output drop of the model's logit for the target class when taking $x_n^{p_m}$, *i.e.*, $x_n$ with patch $p_m$ deleted, as input. As in existing work, we delete a patch by substituting it with a baseline image $b$ of reduced information. In practice, $b$ is often an image of only zeros, random numbers, or $x_n$ blurred with a Gaussian. If not specified otherwise, we use the zero baseline for the remainder of this work (after image normalization). Intuitively, if an attribution method correctly assesses the importance of each patch, the attribution sums will be correlated with the output drops, and accordingly, a high rank correlation implies a better attribution. Formally, we define our in-domain single-deletion score (IDSDS) as

$$\text{IDSDS} := \frac{1}{N} \sum_{n=1}^{N} r\Big( \big\langle \mathcal{A}(f_{t_n}, x_n)_{p_m} \big\rangle, \big\langle f_{t_n}(x_n) - f_{t_n}(x_n^{p_m}) \big\rangle \Big), \tag{1}$$

where $\langle \cdot \rangle \equiv \langle \cdot \mid m = 1, \dots, P \rangle$ denotes an ordered set over index $m$ and $r(\cdot, \cdot) \in [-1, 1]$ is the Spearman rank-order correlation coefficient; here between the attribution sums in each patch $\langle \mathcal{A}(f_{t_n}, x_n)_{p_m} \rangle$ and the model output drops occurring when removing each patch $\langle f_{t_n}(x_n) - f_{t_n}(x_n^{p_m}) \rangle$. Intuitively, the Spearman rank-order correlation coefficient measures the similarity between two rankings $R[X], R[Y]$ of ordered sets $X, Y$. Formally, it can be computed as the Pearson correlation coefficient $\rho$ between the rankings: $r := \rho\big(R[X], R[Y]\big)$.

As described so far, our IDSDS may appear similar to [55]. However, in the following, we go beyond [55] by, first, contributing an *exact* alignment of the train and test domains that is *independent* of the specific attribution method and image label, and therefore, exhibits *no* class information leakage. Second, we exploit the resulting inter-model comparison capabilities for *a novel analysis* of various model designs, including recently proposed intrinsically explainable models.

**Alignment of the train and test domains.** As discussed, an exact alignment of the train and test domains is essential to guarantee that the output changes from image interventions are really due to the removal of image information and not just due to domain shifts. To align the train and test domains for our IDSDS, we substitute one random patch in half of the training images with the considered zero baseline; please refer to Appendix B for a detailed explanation. We only apply this data augmentation to half of the images to ensure that images with *no* deleted patches remain in-domain. Unlike many existing efforts to align the train and test domains when evaluating attribution maps, *e.g.*, [29], our data augmentation is *independent* of the attribution method, and thus, we can use a *single* model to compare *multiple* attribution methods in-domain, enabling a fairer comparison. Further, as our masking is *independent* of the image content and class label, we can guarantee *zero* class information leakage in the masks and, therefore, delete patches *without* adding new information.

*Proof.* Let $H(X)$ be the entropy of a discrete random variable $X$. A binary mask $\boldsymbol{M}$ suffers from *class information leakage* for a class variable $C$ iff the mutual information between the class and the mask $I(C; \boldsymbol{M}) := H(\boldsymbol{M}) - H(\boldsymbol{M}|C)$ is larger than some non-negative "mitigator" $j$ (see [51] for details; the specifics of $j$ are not relevant here), *i.e.*, $I(C; \boldsymbol{M}) > j$. As we specifically design the mask $\boldsymbol{M}$ to be *independent* of the class $C$, we have $H(\boldsymbol{M}|C) = H(\boldsymbol{M})$, hence $I(C; \boldsymbol{M}) = H(\boldsymbol{M}) - H(\boldsymbol{M}) = 0$. With $j$ being non-negative, we know that $I(C; \boldsymbol{M}) \not> j$, and thus, our masking strategy $\boldsymbol{M}$ does not suffer from class information leakage. $\square$

The proof also shows that deleting image content based on more semantic features like image-dependent segmentation masks could introduce information leakage and, thus, is not viable.

**Inter-model comparison.** Unlike the incremental-deletion protocol, the IDSDS can *only* improve if the actual task of ranking the patch importances is more effectively solved and not due to mere changes in the output calibration. Thus, our IDSDS and other single-deletion protocols are an excellent choice for comparing the attribution correctness of *different* models. Interestingly, we found only a few studies that explicitly highlighted or leveraged this opportunity. *E.g.*, in the FunnyBirds framework [28], a part-based single-deletion protocol is used to evaluate the attributions of different models on a *synthetic* dataset. Among the research that considers large-scale datasets like ImageNet [17], such as [55], we are not aware of any work that compares the attributions of *different* models using a

Table 2: *Legend. Raw model-agnostic attribution methods* (●): I×G – Input×Gradient [57], I×G-SG – Input×Gradient & SmoothGrad [61], IG – Integrated Gradients (zero baseline) [64], IG-U – Integrated Gradients (uniform baseline) [63], IG-SG – Integrated Gradients & SmoothGrad. *Absolute model-agnostic methods* (■): "abs." denotes absolute attribution scores, IG-SG-SQ – Integrated Gradients & SmoothGrad squared [29], Saliency [60]. *Perturbation-based methods* (⬠): RISE [47], RISE-U (uniform baseline). *CAM-based and ViT-specific methods* (◆): Grad-CAM [55], Grad-CAM++ [13], SG-CAM++ – Grad-CAM++ & SmoothGrad [42], XGrad-CAM – Axiomatic Grad-CAM [22], Layer-CAM [31], Rollout [1], CheferLRP – Layerwise relevance propagation for ViT [18] as in [14]. *Intrinsically explainable models* (▲): B-cos ResNet-50 [9] and BagNet-33 [7].

| ● I×G | ■ I×G abs. | ■ IG-SG-SQ | ◆ Grad-CAM | ◆ Rollout |
|---|---|---|---|---|
| ● I×G-SG | ■ I×G-SG abs. | ■ Saliency | ◆ Grad-CAM++ | ◆ CheferLRP |
| ● IG | ■ IG abs. | ⬠ RISE | ◆ SG-CAM++ | ▲ B-cos |
| ● IG-U | ■ IG-U abs. | ⬠ RISE-U | ◆ XGrad-CAM | ▲ BagNet-33 |
| ● IG-SG | ■ IG-SG abs. | | ◆ Layer-CAM | |

single-deletion protocol. We here fill this gap and contribute a systematic study of the influence of various design choices on the attribution correctness of ImageNet models in Section 4.3.

**Limitations.** Although our IDSDS has strong theoretical advantages regarding domain alignment, information leakage, and inter-model comparison, it naturally comes with limitations that are important to discuss. First, we lose granularity by evaluating on a patch level and cannot assess the attribution quality *within* each patch. However, this is necessary if we want to avoid two other limitations. If we were to evaluate on a pixel level, we would have to apply interventions on all pixels, respectively, a large number of randomly selected pixels, which significantly increases computation and is not feasible in practice. Alternatively, we could select the pixels depending on the image content to evaluate the most interesting pixels which, however, yields the problem of information leakage. Additionally, in Section 4.1 we show that increasing the number of patches from 16 to 64, effectively increasing the granularity of our evaluation, leads to similar rankings. Another limitation of our IDSDS is the need to fine-tune each model with our proposed data augmentation scheme to align the train and test domains. This increases the computational cost (~7h for a ResNet-50 [25] using our hardware) compared to evaluation protocols that do not require any adjustment of the model, and could alter the model's behavior. However, as we show in Table 3 and Section 4.1, the fine-tuning has almost no negative effect on the classification accuracy and the learned features, and thus, our fine-tuned models are similarly well suited for downstream tasks as the original ones. Finally, as other single-deletion protocols [28, 55], our IDSDS only considers the effect of deleting single patches instead of patch combinations. As discussed in Appendix C, this is reasonable because we *need* to make approximations (the exact explanation is given by the model itself) and there is no single, correct answer as to how simple this approximation should be.

## 4 Experiments

**Preliminaries.** We use the ImageNet-1000 dataset [17, 53]. For networks without fine-tuning (*e.g.*, Figure 2 (b) and (c)), we use pre-trained models [45]. For models where fine-tuning is applied (*cf*. Section 3), we initialize with weights from the pre-trained models and train for 30 epochs with SGD using a weight decay of $1 \times 10^{-4}$, a momentum of 0.9, and a learning rate of 0.001 (0.01 for B-cos ResNet-50 [9]) that is reduced by a factor of 0.1 every ten epochs. Our IDSDS is calculated over the full ImageNet evaluation split. For all of the following experiments, we use a ResNet-50 [25] as a reference. Due to the model's extensive usage in related work, findings will translate to a significant number of existing papers. Further, it comes in many variations, allowing for a granular change of individual design choices to systematically study their effect on the model's attribution correctness. Table 2 shows the legend that applies to all subsequent plots.

Table 3: *ImageNet accuracy (in %), before (pre) and after (post) fine-tuning with our data augmentation, on the regular evaluation split (uncorrupted) and the evaluation split with the patches deleted that result in the lowest accuracy (corrupted).* As our scheme has, at most, a minor negative effect on the accuracy ($\Delta$) of uncorrupted images, it is suitable when evaluating the attribution correctness of ImageNet models.

| Model | Accuracy uncorrupted (%) | | | Accuracy corrupted (%) | | |
|---|---|---|---|---|---|---|
| | pre | post | $\Delta$ | pre | post | $\Delta$ |
| ResNet-18 | 69.76 | 70.55 | +0.79 | 30.84 | 43.42 | +12.58 |
| ResNet-50 | 76.12 | 76.92 | +0.80 | 41.94 | 53.97 | +12.03 |
| ResNet-101 | 77.38 | 77.90 | +0.52 | 45.30 | 55.90 | +10.50 |
| ResNet-152 | 78.32 | 78.78 | +0.46 | 46.91 | 57.52 | +10.61 |
| Wide ResNet-50 | 78.47 | 78.47 | +0.00 | 47.42 | 57.68 | +10.26 |
| ResNet-50 w/o BN | 75.81 | 75.14 | −0.67 | 42.50 | 49.34 | +6.84 |
| ResNet-50 w/o BN w/o bias | 73.51 | 73.28 | −0.23 | 39.46 | 48.05 | +8.59 |
| VGG-16 | 71.59 | 72.07 | +0.48 | 36.39 | 46.31 | +9.92 |
| ViT-B/16 | 81.43 | 82.74 | +1.31 | 57.92 | 64.55 | +6.63 |
| B-cos ResNet-50 | 75.88 | 75.81 | −0.07 | 34.13 | 38.26 | +4.13 |
| BagNet-33 | 64.21 | 68.37 | +4.16 | 47.36 | 55.21 | +7.85 |

## 4.1 Sanity testing our IDSDS

We start by verifying if our assumptions and used hyperparameters are sensible, respectively what effect they have on the final rankings. Due to space limitations, we include the accompanying figures in Appendix A and report the corresponding Spearman rank-order correlations in the main text.

**Accuracy.** Since our IDSDS requires fine-tuning, we cannot directly evaluate existing networks. To use our fine-tuned models for the same downstream tasks as the original models, it is thus important that the classification accuracy does not suffer. In Table 3, we compare the top-1 accuracy on the uncorrupted ImageNet evaluation split for pre-trained networks and models fine-tuned with our data augmentation. Our fine-tuning is only a minor adjustment that has almost no negative effect on the evaluation accuracy of the examined models, and hence, can be used without hesitation. Further, we evaluate the accuracy of images with deleted patches to verify if the train and test domains have been aligned as intended. For each image, we choose the patch that results in the lowest accuracy, *i.e.*, we select the "worst-case" patch for deletion. Our fine-tuned models consistently outperform the regular models on corrupted samples, indicating a better alignment of the train and test domains. The differences in accuracy between the uncorrupted and the corrupted images can be ascribed to important information being removed, which makes it harder to correctly classify the image at hand.

**Network similarity.** To further ensure that the original model (OOD) and the fine-tuned model (ID) behave similarly, we conduct the following three experiments. First, we measure the mean absolute difference (MAD) between the target softmax outputs of the two models. We suspect that a model using different features will result in different output confidences. Thus, a smaller value indicates more similar models. The MAD between the target softmax outputs for the original ResNet-50 (OOD) and our fine-tuned ResNet-50 (ID) is 0.049. Between an OOD and ID VGG-16 we obtain a score of 0.028. As a comparison, the OOD ResNet-50 and the OOD VGG-16 have a much higher MAD of 0.133. Second, we measure the mean absolute difference between the GradCAM attributions of the two models. We chose GradCAM for its simplicity and because it is less noisy than many other attribution methods. Intuitively, similar models should yield similar attribution maps, and thus, a smaller value again indicates more similar models. We compare the same models as before and obtain scores of 0.047 for the original ResNet-50 (OOD) and our fine-tuned ResNet-50 (ID), 0.039 for the OOD VGG-16 and the ID VGG-16, and again a much higher score of 0.193 for the OOD ResNet-50 and the OOD VGG-16. Third, we randomly select channels from the last convolutional layer of the models and plot the images that result in the highest activation of that channel. This is an established method to visualize the "concept(s)" learned by a channel, and for similar models, the shown images should be similar. Results can be found in Appendix A. The selected channels react to more or less the same images for the original OOD model and our fine-tuned ID model. To conclude, in both our quantitative evaluations, the fine-tuned models are significantly more similar to the corresponding OOD model than the two different OOD baseline models (ResNet-50/VGG-16). Additionally, the

qualitative analysis shows that randomly selected channels react to more or less the same images for the OOD and ID models. These results are a strong indicator that our ID models still use very similar features as the original OOD models (besides that they learned to better predict images with deleted patches), and thus, our proposed method is further validated.

**Patch number.** We measure the influence of the number of patches $P$ on our IDSDS in Appendix A. With fewer patches, ranking the importance of each patch is easier, resulting in a higher IDSDS. Conversely, with more patches, the task is harder, and the IDSDS is lower. When $P$ is too large (*e.g.*, 64), most methods struggle, clustering around an IDSDS of 0 to 0.06, making the IDSDS no longer sufficiently discriminative. Interestingly, for smaller $P$, coarser CAM-based methods perform slightly better, possibly due to their lower resolution. Nonetheless, the high Spearman rank-order correlations (0.93 between $P = 4$ and $P = 16$; 0.9 between $P = 16$ and $P = 64$) indicate stable rankings across different $P$, which is another advantage of our IDSDS. In our experiments with ImageNet images, we found $P = 16$ to produce results in a useful range. However, if attribution methods improve in the future, one can increase $P$ to adjust the evaluation protocol.

**Baseline.** A common limitation of deletion-based protocols is their dependence on the used baseline image [65]. To evaluate how sensitive our proposed IDSDS is to different baseline images, we measure the effect of fine-tuning and evaluating with the zero baseline, with random baseline images (values drawn uniformly in $(-1, 1)$), and with blurry baseline images ($51 \times 51$ Gaussian, $\sigma = 41$) in Appendix A. The ranking from the zero baseline has a rank correlation of 0.99 with the random baseline ranking and 0.96 with the blurry baseline ranking. Thus, our protocol is very stable under changing baseline images, which is another strong advantage for drawing clearer conclusions.

**Stability.** To ensure that our results are conclusive and stable under differing training runs, we further compare our IDSDS for different training seeds in Appendix A. The scores, as well as the ranking, are extremely stable (rank correlation $\geq$ 0.996 between all seeds). To be mindful of our energy consumption, we report the results from a single model with a seed of zero in all other plots.

## 4.2 Ranking attribution methods

Now that we have established a theoretically and empirically sound evaluation protocol, we study how different attribution methods perform in our proposed IDSDS on the ImageNet dataset with a ResNet-50 in Figure 2 (a). Integrated Gradients with a uniform baseline (IG-U) [63] performs the best among all examined model-agnostic attribution methods. Surprisingly, it even outperforms Integrated Gradients (IG) [64], despite IG using the same zero baseline as our patch interventions. The opposite holds for RISE where a black baseline achieves better results than the uniform baseline (RISE-U). Generally, CAM-based methods perform quite well, with Grad-CAM [55] and Axiomatic Grad-CAM (XGrad-CAM) [22] performing the best. SmoothGrad (SG) [61] impairs the performance for all, Smooth Grad-CAM++ (SG-CAM++) [42], I×G with SG (I×G-SG), and IG with SG (IG-SG), indicating that it reduces the correctness of attribution maps, which is in line with findings of Hooker et al. [29]. Intriguingly, taking the absolute value of the attribution maps hurts correctness for the better performing methods (*e.g.*, IG-U *vs*. IG-U abs.), which is in stark contrast to findings from related work (see below). Finally, even the best-performing model-agnostic method only achieves an IDSDS of 0.25, which highlights the need for attribution methods with higher correctness. The intrinsically explainable models B-cos ResNet-50 [9] and BagNet-33 [7] achieve a significantly higher IDSDS than standard attribution methods on the ResNet-50 backbone, with BagNet-33 achieving an astonishing IDSDS of 0.797. Thus, researching intrinsically explainable models is a promising research direction to which we contribute by examining the effect of different model designs on attribution correctness in Section 4.3.

**Absolute attributions.** Intrigued by our finding of raw attribution values outperforming absolute ones (*cf*. Figure 2 (a)), yet contrasting findings in related work [62, 66], we will now investigate this issue. To this end, we take a closer look at the protocol that was used to establish these insights in prior work, *i.e.*, the *incremental-deletion score* (IDS) (see Section 2). In most previous work, the attribution used is *fixed* and computed for the original input image without any pixels removed. However, it is also sensible to compute *updated* attributions based on images with removed pixels after *each* degradation step [39]. Although not the standard way of computing IDS, this allows varying the amount of deleted information between the image for which the attribution has been computed and the intervened image. In the updated attribution setup, this amount is very small. In the fixed attribution setup, this amount can be very high as the intervened image can have almost all pixels deleted.

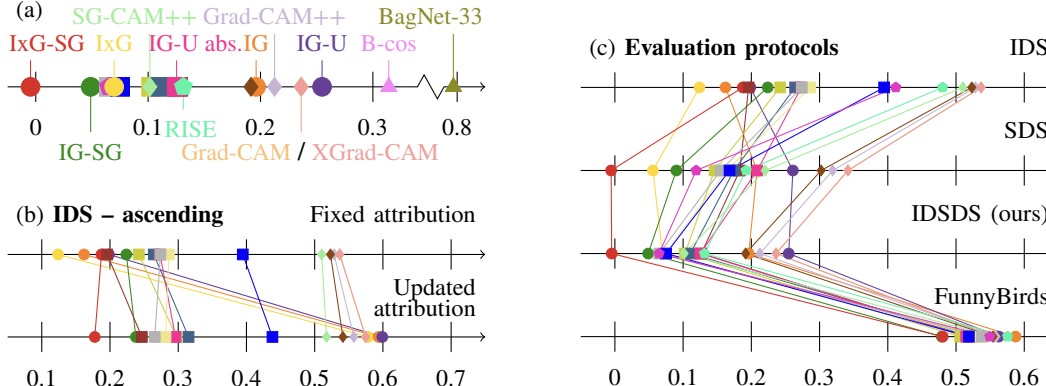

Figure 2: *(a) IDSDS on ImageNet for attribution methods using a ResNet-50 and the considered intrinsically explainable models.* Please refer to Section 4.2 for an interpretation of the results. *(b) Comparison of the incremental-deletion score (IDS) when computing a fixed attribution for the original input* (top) *versus when updating the attribution in each deletion step* (bottom). The raw attributions for IxG, IG, and IG-U perform better for the second setup. *(c) Comparison of existing evaluation protocols.* We compare IDSDS to the incremental-deletion protocol [54] (IDS), the OOD single-deletion protocol [55] (SDS), and FunnyBirds [28]. Notably, the change between SDS and IDSDS indicates that aligning the training and testing domains is important; IDS is the only protocol strictly preferring absolute over raw attributions, and the best baseline image changes between real images and synthetic images from FunnyBirds. For better readability, we provide numerical values in Appendix F.

In Figure 2 (b), we compare how these two ways of computing the attribution affect the IDS. For the fixed attribution, we confirm prior findings that raw attributions perform worse than absolute ones. When updating the attribution at each degradation step, we see a gain in IDS for the raw attribution values of IxG, IG, and IG-U, making them the best-performing methods. This might be due to the amount of deleted information between the intervened and attributed images. For larger amounts, the magnitude of attributions seems more relevant, favoring absolute values. For smaller amounts, the sign becomes more important for attribution correctness. Since images with one deleted patch in our IDSDS are also fairly similar to the original image for which the attribution was computed, this could explain why raw attribution values also outperform absolute attributions in our IDSDS.

**Comparison to related work.** To get a better understanding of the distinctions between existing deletion-based protocols, in Figure 2 (c), we compare our IDSDS with three well-established protocols: the incremental-deletion score [54] (IDS), the OOD single-deletion score [55] (SDS), and the single deletion protocol in FunnyBirds [28]. If applicable, we use the same hyperparameters, such as baseline image and number of patches, to ensure a fair comparison. Therefore, the difference between the SDS and our IDSDS boils down to having unaligned *vs.* aligned train and test domains.

The IDS is the only protocol where absolute attribution values are strictly preferred over raw values, supporting our findings in Section 4.2. CAM-based approaches outperform competing methods for protocols with OOD issues (IDS and SDS). For protocols with aligned train and test domains, IG or IG-U perform best. Interestingly, the preferred baseline image changes between synthetic (black, FunnyBirds) and real images (uniform, IDSDS), indicating dataset dependence. While SDS and IDSDS have a fairly high Spearman rank-order correlation of 0.89, we observe interesting ranking changes when aligning train and test domains. This, along with the theoretical advantages of alignment [29], underscores the importance of fine-tuning with our data augmentation. As evaluating the quality of evaluation protocols is challenging, we believe that striving for theoretical guarantees, such as aligned domains, is a crucial step toward faithfully evaluating attribution methods. Further, each protocol measures a slightly different proxy for attribution quality, and thus, including multiple protocols such as done in Quantus [26] or FunnyBirds [28] is a promising direction for a more granular evaluation.

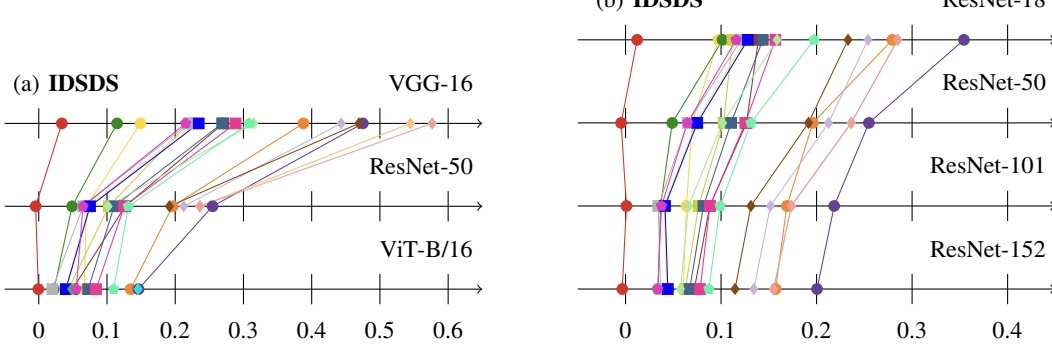

Figure 3: *(a) Comparison of model architectures (VGG-16 [59], ResNet-50 [25], and ViT-B/16 [18]). Compared to ResNet-50, attribution methods achieve a higher IDSDS on VGG-16 and a lower IDSDS on ViT-B/16. (b) Comparison of network depths.* The IDSDS decreases with increasing depth.

### 4.3 How design choices affect attribution correctness

We conclude our experiments by studying how the model design affects attribution correctness. To do so, we compare various different attribution methods on multiple setups (*e.g.*, different models). We say that the attribution correctness of a model increases if the IDSDS of the majority of attribution methods increases for that model. Please note that this phrasing is slightly imprecise because we only consider a subset of *all* attribution methods (albeit a large one). However, with very clear tendencies becoming visible in our results, we argue that this colloquial phrasing is tolerable. We here focus on ResNet models but provide a similar analysis for VGG [59] in Appendix D, confirming our findings. Whilst some of our insights align with intuition and thus may not be too surprising, we are not aware of any work that examines the following aspects for ImageNet models in such a systematic manner and for such a variety of attribution methods. We believe that our findings are highly relevant for the XAI community and for applications where explainability is crucial. Specifically, we show what design choices are well suited for achieving the highest quality attributions, and we provide the first work that empirically confirms the accuracy-explainability trade-off [6, 40] in a large-scale study.

**Architecture.** We measure how the IDSDS for different attribution methods changes across different backbones in Figure 3 (a). The backbones produce drastically different results, with VGG-16 [59] exhibiting more correct attributions than ViT-B/16 [18]. The rankings of the examined methods remain fairly stable over all backbones. For the ViT-specific methods, Rollout is ranked in the middle, while CheferLRP achieves, together with IG-U, the best IDSDS on ViT-B/16.

**Depth & width.** More parameters could lead to more complex and less correctly attributable models. To verify this, we measure the IDSDS on four ResNet models with increasing depths of 18, 50, 101, and 152 layers in Figure 3 (b). Confirming our assumption, the IDSDS, and thus the attribution correctness, decreases with increasing depths. In Figure 4 (a), we compare the IDSDS between a standard ResNet-50 and a wide ResNet-50 [67]. Again, the IDSDS for most attribution methods decreases when using the wider W-ResNet-50 network, indicating that a larger width impairs the attribution correctness of the model.

**Bias term & BN.** Since batch normalization (BN) [30] and bias terms can be removed from DNNs without losing significant accuracy [27, 69], we study how this affects our IDSDS in Figure 5 (a). Without BN layers [69], there is a clear improvement in IDSDS for all examined methods. When additionally removing the remaining bias terms, the IDSDS for almost all methods improves even further. As theoretically established in previous studies [27, 37], our IDSDS empirically confirms that removing the bias term has a positive effect on attribution correctness. The positive impact of removing only BN layers is a new discovery, potentially linked to the partial removal of bias terms or the negative influence of normalization layers themselves.

**Softmax.** Depending on the implementation, the final softmax layer of a classification model can either be part of the model or the loss. Thus, attributions can also be computed w.r.t. the pre- or post-softmax output. While Lerma and Lucas [35] discussed this issue and the resulting implications for attribution maps from a theoretical perspective, we provide a quantitative comparison in Figure 4 (b).

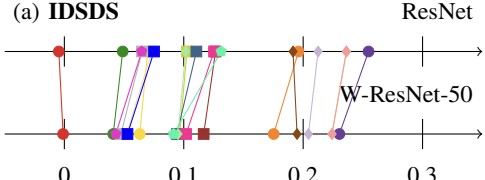
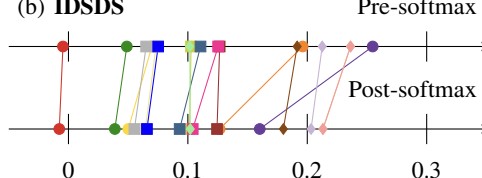

Figure 4: *(a) Comparison of different widths.* Almost all attribution methods achieve a lower IDSDS for the wide (W) ResNet-50 [67], indicating that the increased width impedes attribution correctness. *(b) Comparison of pre- and post-softmax attribution maps.* Computing the attribution for a ResNet-50 after the final softmax layer reduces the IDSDS of almost every attribution method.

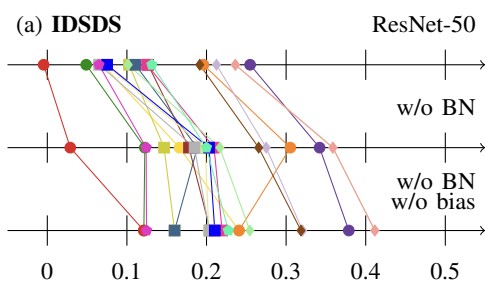
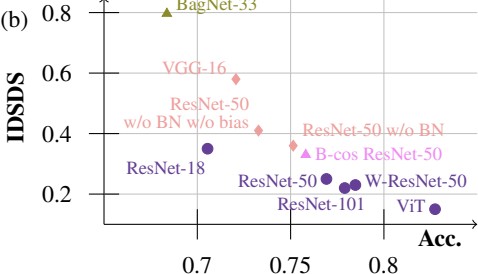

Figure 5: *(a) Comparison of batch norm (BN) and the bias term.* Removing the BN layers and all bias terms positively affects the IDSDS. *(b) IDSDS over accuracy.* We plot the best IDSDS of each model over the top-1 ImageNet accuracy. The mark indicates the respective best attribution method.

Computing the attributions after the final softmax layer reduces the correctness as measured by our IDSDS for almost all methods, indicating that pre-softmax attributions are favorable.

**IDSDS *vs*. accuracy.** We conclude our analysis by plotting the best IDSDS of each model over the top-1 ImageNet accuracy in Figure 5 (b). The accuracy and IDSDS appear to be anticorrelated, empirically supporting the hypothesis of the often-mentioned accuracy-explainability trade-off [6, 40]. However, certain architectural changes favor this trade-off more than others. For example, the accuracy gain obtained by increasing the depth of the network comes with a higher IDSDS drop than when increasing the width of the network. Further, there is a tendency that for more correctly attributable models XGrad-CAM (♦) and the similarly performing Grad-CAM are preferable, while IG-U (●) produces the most correct attributions for the less correctly attributable models. We hypothesize that for the less correctly attributable models, it is important to consider the full network as is done by IG-U. On the other hand, the more correctly attributable models may be simple enough so that focusing on the last layer as done in Grad-CAM suffices to produce correct attributions.

## 5 Conclusion

We propose a novel in-domain single-deletion score (IDSDS) that overcomes two major limitations of existing protocols for evaluating attribution correctness: the OOD issue (respectively, information leakage) and lacking inter-model comparisons. Using our IDSDS to rank 23 attribution methods, we find that intrinsically explainable models outperform standard models by a large margin, that Integrated Gradients can surpass CAM-based and perturbation-based methods, and that the sign of attribution values is more important than previously assumed. Additionally, we measure the influence of different model design choices on the attribution quality of ImageNet models. We discover that some design choices consistently improve attribution correctness for a wide range of attribution methods, that there is an accuracy-IDSDS trade-off, and that some choices favor this trade-off more than others, which we hope will facilitate the future development of more explainable models.

## Acknowledgments

This project has received funding from the European Research Council (ERC) under the European Union's Horizon 2020 research and innovation programme (grant agreement No. 866008). The project has also been supported in part by the State of Hesse through the cluster projects "The Third Wave of Artificial Intelligence (3AI)" and "The Adaptive Mind (TAM)".

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

# A    Sanity testing our IDSDS

As discussed in Section 4.1, we provide additional figures measuring how different numbers of patches (Figure 6 (a)), different baseline images (Figure 6 (b)), and different training seeds (Figure 7) affect our proposed in-domain single-deletion score (IDSDS). Additionally, in Figure 10 we compare randomly selected channels for two models and our corresponding fine-tuned models. For an interpretation of the results, please refer to Section 4.1.

# B    Why our data-augmentation scheme aligns the training and testing domains

To align the train and test domains, we train on images with either no deleted patches or one patch deleted. More formally, we first assume that for the original ImageNet dataset, the train and evaluation domains are aligned (while this assumption may be debatable, it is established consensus within the community). To follow our argument, let us assume that we sample a sufficiently large number of images from our proposed training set (with our data augmentation, see Section 3). For each image that we sample during training, we randomly delete one of the $P = 16$ patches with a probability of 0.5. The other sampled images are left in their original state. If we now sample a very large number of images (the same image can be sampled several times), we, therefore, have a ratio of $16 : 1 : 1 : \cdots : 1$ between the original, uncorrupted images and images where patch $p_m$ with

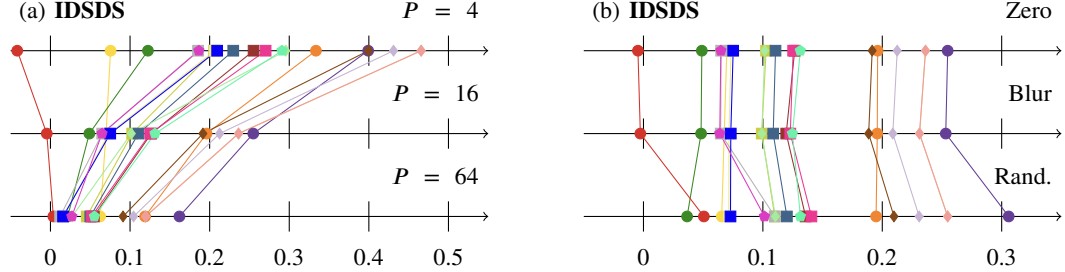

Figure 6: *(a) Comparison of different numbers of patches P*. We measure the IDSDS when using 4, 16, and 64 patches for our IDSDS. The rankings for $P = 4$ and $P = 16$ have a rank correlation coefficient of 0.93, and the rankings for $P = 16$ and $P = 64$ have a rank correlation coefficient of 0.9. Thus, the rankings are quite stable under different $P$. *(b) Comparison of different baselines.* We compare three different kinds of baseline images (zero, blur, and random). The ranking is only slightly affected by the different baselines, showing that our protocol is quite stable w.r.t. the used baseline.

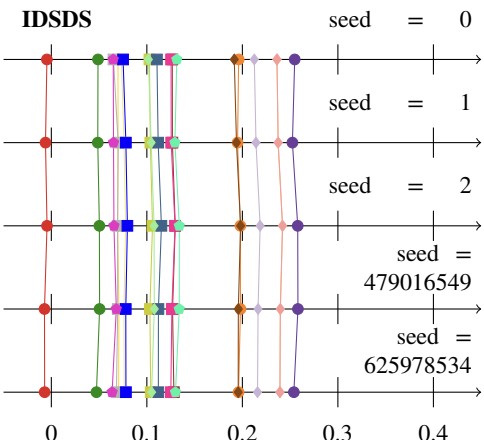

Figure 7: *Comparison of different training seeds.* To verify if our results are stable and conclusive, we compare the results for five different ResNet-50 models fine-tuned with varying seeds. The IDSDS is almost unchanged with different training seeds.

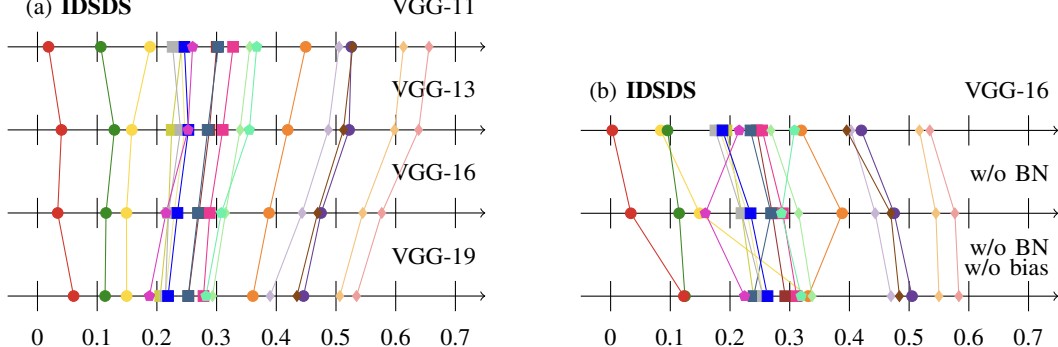

Figure 8: *(a) Comparison of different network depths.* The IDSDS for VGGs with increasing depths decreases. *(b) Comparison of batch norm (BN) and the bias term.* We evaluate how removing the BN layers and all bias terms affects the IDSDS in a VGG-16 network. Both modifications positively affect the IDSDS for almost all attribution methods.

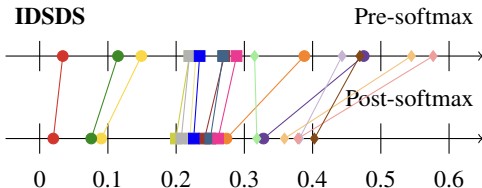

Figure 9: *Comparison of pre- and post-softmax attribution maps.* Computing the attribution for a VGG-16 network after the final softmax layer reduces the IDSDS of almost every method.

$m \in [0, \dots, 15]$ is deleted. At test time, for each image, we compare the output of the model for the original image with the output of the model when one of the $P = 16$ patches is deleted. So for each test image, we have (for $P = 16$) exactly 16 forward passes for the original image (as the results are the same for those, we only compute one forward pass in practice) and one for each of the deleted patches, which results in a ratio of $16 : 1 : 1 : \cdots : 1$ that corresponds exactly to the ratio used in the training domain. To conclude, we evaluate and train on both uncorrupted images and images with exactly one patch deleted, maintaining the same sampling probability at train and test time.

## C  Why interventions are reasonable for assessing feature importance

In deletion-based protocols, we perform image interventions to generate target importance scores. This is reasonable as explanations aim to approximate the model's causal structure, and because the causal structure can be estimated via interventions [28, 46]. Considering that the model itself already yields the *true* causal structure that, however, is too complex to understand, a *simplified approximation* is sought, instead. Consequently, deletion-based protocols assume such simplified approximations. *E.g.*, single-deletion protocols assume that such a simplified model processes each feature independently, which is not necessarily true for the *real* model. As similar simplifications are implicitly made in all existing deletion-based protocols, we regard this circumstance as given and only mention it here for completeness.

## D  VGG results

To ensure that our findings in the main paper do not only apply to ResNets [25], we additionally test how network depths, batch normalization (BN) layers, bias terms, and pre-/post-softmax attributions affect the attribution correctness of VGG models [59] in Figures 8 (a) and (b) and Figure 9. Confirming our findings for ResNets in the main paper, increasing the depth decreases the attribution correctness as measured by our IDSDS, removing the BN layers and bias terms increases the attribution correctness, and using pre-softmax attributions results in higher IDSDS. From this, we can conclude that our findings generalize beyond ResNets.

Table 4: *Numerical results for our plot in Figure 2 (a) and (c) – IDSDS.*

| Method | IDSDS | Method | IDSDS |
|---|---|---|---|
| I×G | 0.07 | Saliency | 0.127 |
| I×G-SG | -0.005 | RISE | 0.131 |
| IG | 0.196 | RISE-U | 0.065 |
| IG-U | 0.255 | Grad-CAM | 0.236 |
| IG-SG | 0.049 | Grad-CAM++ | 0.213 |
| I×G abs. | 0.103 | SG-CAM++ | 0.102 |
| I×G-SG abs. | 0.065 | XGrad-CAM | 0.236 |
| IG abs. | 0.111 | Layer-CAM | 0.192 |
| IG-U abs. | 0.125 | B-cos | 0.314 |
| IG-SG abs. | 0.073 | BagNet | 0.797 |
| IG-SG-SQ | 0.075 | | |

Table 5: *Numerical results for our plot in Figure 2 (c) – IDS.*

| Method | IDS | Method | IDS |
|---|---|---|---|
| I×G | 0.124 | IG-SG-SQ | 0.395 |
| I×G-SG | 0.188 | Saliency | 0.196 |
| IG | 0.162 | RISE | 0.48 |
| IG-U | 0.198 | RISE-U | 0.412 |
| IG-SG | 0.224 | Grad-CAM | 0.537 |
| I×G abs. | 0.242 | Grad-CAM++ | 0.528 |
| I×G-SG abs. | 0.274 | SG-CAM++ | 0.510 |
| IG abs. | 0.264 | XGrad-CAM | 0.537 |
| IG-U abs. | 0.273 | Layer-CAM | 0.523 |
| IG-SG abs. | 0.286 | | |

# E    Experimental details

When fine-tuning models with our proposed data-augmentation scheme (*cf*. Section 3), we initialize with weights from pre-trained ImageNet models and train for 30 epochs with SGD using a weight decay of $1 \times 10^{-4}$, a momentum of 0.9, and a learning rate of 0.001 (0.01 for B-cos ResNet-50 [9]) that is reduced by a factor of 0.1 every ten epochs. We use a batch size of 256 for all models. We use servers with up to four NVIDIA A100-SXM4 (40GB), NVIDIA RTX A6000 (48GB), or NVIDIA RTX 6000 Ada (48GB) GPUs. The code is implemented in PyTorch [45] published under a 3-Clause BSD License. For most attribution methods, we use Captum [33] (3-Clause BSD License). For CAM-based methods, we use TorchCAM [20] (Apache 2.0 License). The RISE implementation is taken from [47] (MIT License). Implementations for Rollout and CheferLRP are from [14] (MIT License). BagNet [7] is available under the MIT License, and the B-cos network [11] under the Apache 2.0 License.

For the incremental-deletion score, we use 32 degradation steps. We use the zero baseline as done in our IDSDS.

# F    Numerical results

As it is difficult to read off the *exact* numbers from the plots in the main paper, we report the results for Figures 2 (a) and (c) in numerical fashion in Tables 4 to 7. For the other experiments, the *comparison* between the different setups is the focus of our work, which is why we emphasize the changes in ranking visible from the plots rather than any absolute numbers.

Table 6: *Numerical results for our plot in Figure 2 (c) – SDS.*

| Method | SDS | Method | SDS |
|---|---|---|---|
| I×G | 0.056 | IG-SG-SQ | 0.168 |
| I×G-SG | -0.006 | Saliency | 0.183 |
| IG | 0.208 | RISE | 0.192 |
| IG-U | 0.261 | RISE-U | 0.119 |
| IG-SG | 0.09 | Grad-CAM | 0.342 |
| I×G abs. | 0.146 | Grad-CAM++ | 0.319 |
| I×G-SG abs. | 0.155 | SG-CAM++ | 0.22 |
| IG abs. | 0.174 | XGrad-CAM | 0.342 |
| IG-U abs. | 0.209 | Layer-CAM | 0.302 |
| IG-SG abs. | 0.164 | | |

Table 7: *Numerical results for our plot in Figure 2 (c) – FunnyBirds.*

| Method | FunnyBirds | Method | FunnyBirds |
|---|---|---|---|
| I×G | 0.545 | IG-SG-SQ | 0.519 |
| I×G-SG | 0.48 | Saliency | 0.517 |
| IG | 0.587 | RISE | 0.576 |
| IG-U | 0.562 | RISE-U | 0.549 |
| IG-SG | 0.479 | Grad-CAM | 0.555 |
| I×G abs. | 0.506 | Grad-CAM++ | 0.559 |
| I×G-SG abs. | 0.538 | SG-CAM++ | 0.555 |
| IG abs. | 0.523 | XGrad-CAM | 0.555 |
| IG-U abs. | 0.515 | Layer-CAM | 0.554 |
| IG-SG abs. | 0.532 | | |

| Model | Channel | |
|-------|---------|---|
| OOD ResNet-50 | 15 |  |
| ID ResNet-50 | 15 |  |
| OOD ResNet-50 | 106 |  |
| ID ResNet-50 | 106 |  |
| OOD ResNet-50 | 676 |  |
| ID ResNet-50 | 676 |  |
| OOD VGG-16 | 129 |  |
| ID VGG-16 | 129 |  |
| OOD VGG-16 | 407 |  |
| ID VGG-16 | 407 |  |
| OOD VGG-16 | 495 |  |
| ID VGG-16 | 495 |  |

Figure 10: *Highest activating images for different channels in the last convolutional layer of a ResNet-50 [25] and a VGG-16 [59] (ODD and ID).* The highest activating images are quite similar for the same channel of the OOD and ID models, indicating that the two models behave similarly.

