# OpenReview forum: "Benchmarking the Attribution Quality of Vision Models"
_NeurIPS.cc/2024/Datasets_and_Benchmarks_Track — NeurIPS 2024 Track Datasets and Benchmarks Poster_

### Official Review · Reviewer_tUJa · 2024-07-02
**New attribution metric that advances the field**

**Rating:** 7
**Confidence:** 4
**Clarity:** The paper is very well written and pr…

**Review:**

This paper presents a new attribution evaluation methodology. The authors specifically target the
deletion-based metric which removes pixels from an image based on the attributions for the pixel
to measure the effect on the model output. They properly identify and address the well-identified
problems with deletion: the image may be pushed OOD leading to an unreliable evaluation and
the deletion method cannot compare across different models. They clearly and wholly address
and solve both of these issues through their method. Using the proper experimentation, they
verify the improvements of their method over existing methods. Further experimentation yields
interesting findings for the field. They make deductions about the value of different popular
attribution methods across multiple models, address how specific architecture features effect a
model’s interpretability, and find empirical evidence for the accuracy-explainability tradeoff.

**Strengths:**

1. An interesting and novel solution is presented to the problems they address: creating
OOD data under deletion and the lack of inter-model comparison of existing attribution
metrics.

2. The authors present extensive and proper evaluation of their method both in IRR and ICR
testing for rating reliability and reliability under baseline modification.

3. Further analysis of many attribution methods and the effects of changes to model
architecture add interesting findings to this paper's strong main contribution. The
experimental backing of the accuracy-explainability tradeoff is also a valuable
contribution to the literature.

**Additional Feedback:**

n/a

**Correctness:**

The presentation of the work is correct in my interpretation. The experiments are also properly
performed and were chosen properly for the domain.

**Documentation:**

I have confidence that with some effort I could replicate this work.  Given code is provided in tandem with the text, I see no issues with replicating this work

**Ethics:**

No concerns.

**Limitations:**

I believe the authors thoroughly and wholly addressed the limitations of this work. The only unstated (but
inherent) limitation is the increased evaluation time as a result of the fine-tuning. However, there exist
multiple widely-accepted methods which also perform fine-tuning, making this a minor issue that does
not in any meaningful way outweigh the positives of this method.

**Opportunities For Improvement:**

I do not see much evident room for improvement with the method itself as presented. My only
recommendations could be related to presentation. While everything was clearly presented, I think the
presentation would benefit from increased usage of formal definitions. For example, an additional line describing the Spear-man rank order coefficient would be useful. In addition to this, more time could be spent addressing related work which is unexplored (methods that rely
on ground truth, other methods besides ROAR which use training), but this does not make me doubt the  contributions or knowledge of the authors as the paper is presented.

**Relation To Prior Work:**

The differences of this work from prior work as well as the similarities are clearly outlined. The authors
are clearly aware of the field.

**Summary And Contributions:**

The authors present an attribution quality metric that advances the SOTA. They remedy a few long-standing problems within the field such as (i) alignment of train/test domains and (ii) inter-model comparison. In addition to this, the experimentation of their method
creates new findings in the field and proves their method is effective.

---

> ### Author Rebuttal · Authors · 2024-08-16
>
> We are thankful for the reviewer's effort, useful suggestions, and for highlighting that the paper provides a strong main contribution and that the experimental backing of the accuracy-explainability tradeoff is a valuable contribution. We address the concerns below and will revise the paper accordingly.
>
> > "While everything was clearly presented, I think the presentation would benefit from increased usage of formal definitions. For example, an additional line describing the Spear-man rank order coefficient would be useful. In addition to this, more time could be spent addressing related work which is unexplored (methods that rely on ground truth, other methods besides ROAR which use training), but this does not make me doubt the contributions or knowledge of the authors as the paper is presented."
>
> Due to the space limitation, we unfortunately had to shorten a few formal definitions and related work. With the additional page provided in the case of acceptance, we will gladly add more precise formal definitions (e.g., of the correlation measure; see also response to Reviewer einR) and will expand the discussion of related work to further improve the quality of our work.
>
> > "I believe the authors thoroughly and wholly addressed the limitations of this work. The only unstated (but inherent) limitation is the increased evaluation time as a result of the fine-tuning. However, there exist multiple widely-accepted methods which also perform fine-tuning, making this a minor issue that does not in any meaningful way outweigh the positives of this method."
>
> We thank the reviewer for pointing out this additional limitation. We will add it to the limitations section of our work. While our method certainly has a higher run-time than approaches that do not require fine-tuning, we would like to add that we are still orders of magnitude more efficient than ROAR, which requires training per corruption level and attribution method. We only require a single fine-tuning of each model.
>
> We again thank the reviewer and look forward to further discussion.

---

### Official Review · Reviewer_einR · 2024-07-20
**review for 704**

**Rating:** 6
**Confidence:** 3
**Correctness:** yes
**Clarity:** yes

**Review:**

Overall, the paper's methodological advancements and empirical findings make it
a noteworthy contribution to explainable AI in computer vision.

**Strengths:**

1. The proposed evaluation protocol allows for fair comparisons between different attribution methods and avoids misalignment between training and testing domains.

2. The paper comprehensively evaluates a wide range of attribution methods (23 in total) and examines the effects of eight different design choices of vision models on attribution quality.

3. The empirical findings demonstrate that intrinsically explainable models outperform standard models in terms of attribution quality, challenging previous findings. This provides valuable insights for future model design and interpretability research.

**Additional Feedback:**

cf. Opportunities For Improvement

**Documentation:**

yes

**Limitations:**

yes

**Opportunities For Improvement:**

1. As stated by the authors, the IDSDS protocol loses granularity by evaluating
the patch level rather than the pixel level. This trade-off is necessary for
computational feasibility, as stated by authors, but could limit the
assessment's precision.

2. The protocol requires fine-tuning the model and this may alter the model's
behavior. As shown in Table 3, the changes in `Accuracy uncorrupted` is
insignificant, but the changes in `Accuracy corrupted` should not be ignored.

3. Some other:
  The description and explanation of Figure 2 (c) are not clear for understanding
    the advantages of the proposed protocol compared to the existing ones.
  - The definition of `correlation` is not clear. For example, common metrics
    like Pearson correlation, Spearman correlation, or Kendall tau correlation.
  - The variations of training seeds (Figure 7 (a)) and network depth
    (Figure 8 (a)) are not varied enough to make a strong conclusion.

**Relation To Prior Work:**

yes

**Summary And Contributions:**

This paper addresses the evaluation of attribution methods used to explain the
predictions of DNNs in computer vision. The authors propose an evaluation
protocol called the In-Domain Single-Deletion Score (IDSDS), which overcomes
limitations such as out-of-domain issues and the inability to compare different
models.

---

> ### Author Rebuttal · Authors · 2024-08-16
>
> We appreciate the reviewer's effort, insightful feedback, and that they consider our work to add valuable insights for future model design and interpretability research. We will address the key concerns outlined below and revise the paper accordingly.
>
> > "As stated by the authors, the IDSDS protocol loses granularity by evaluating the patch level rather than the pixel level. This trade-off is necessary for computational feasibility, as stated by authors, but could limit the assessment's precision."
>
> We fully agree with this point and recognize it as a valid limitation of our work (l. 156). To add to this point, we would like to refer to the discussion in lines 197-206 and note that we show in Figure 6 (a) in the Appendix that increasing the number of patches from 16 to 64, effectively increasing the granularity of our evaluation, leads to similar rankings. We will include this insight more explicitly in the limitations section.
>
> > "As shown in Table 3, the changes in Accuracy uncorrupted is insignificant, but the changes in Accuracy corrupted should not be ignored."
>
> We agree that this should not be ignored and we acknowledge that the models do, in fact, learn new or additional features when fine-tuning with our data augmentation. The intent behind Table 3 and the nearly unchanged accuracy on uncorrupted images was to emphasize that if a user wants to ensure their model is evaluated directly using our IDSDS, they could simply employ the fine-tuned model for their downstream tasks. However, we will include a discussion, noting that the improved accuracy on corrupted images suggests that the models are learning at least slightly different features.
>
> > "The description and explanation of Figure 2 (c) are not clear for understanding the advantages of the proposed protocol compared to the existing ones."
>
> We agree with the reviewer and will revise the caption accordingly. For a detailed description, please refer to lines 257-274 of the main paper. In short, (1) we see a change between SDS and IDSDS, indicating that aligning the training and testing domains is important, (2) IDS is the only protocol strictly preferring absolute attributions, and (3) the best baseline changes between real images and synthetic images (from FunnyBirds).
>
> > "The definition of correlation is not clear. For example, common metrics like Pearson correlation, Spearman correlation, or Kendall tau correlation."
>
> We apologize that this point was not sufficiently clear. While we mention that we use the Spearman rank-order correlation coefficient in line 116, we understand that this is easily overlooked and should be highlighted more clearly -- we will revise the paper accordingly.
>
> > "The variations of training seeds (Figure 7 (a)) and network depth (Figure 8 (a)) are not varied enough to make a strong conclusion."
>
> To include more empirical evidence and reinforce our findings, we ran additional experiments with two more randomly chosen seeds (479016549 and 625978534), giving us a total of five different seeds, and with two more network depths (ResNet-152 and VGG-11). The updated figures (Figure 7 (a), Figure 8 (a), and Figure 3(b)) are included in the attached PDF. The additional experiments confirm our findings from the initial experiments, i.e., increasing the network depth decreases the attribution quality and our proposed IDSDS is extremely stable under different training seeds.
>
> We again thank the reviewer and look forward to further discussion.

---

> > ### Author Response · Authors · 2024-08-26
> > **Follow-up on “Opportunities For Improvement - 2“**
> >
> > Reviewer beW6 has raised a similar point as “Opportunities For Improvement - 2“ and proposed to include an experiment to measure the similarity between the fine-tuned and original models. We have implemented the experiment and report the results at https://openreview.net/forum?id=XmyxQaTyck&noteId=cOsb9MUPYj (Q2). To summarize, we can quantitatively and qualitatively show that the fine-tuned models behave very similarly to the original model.

---

### Official Review · Reviewer_beW6 · 2024-07-25

**Rating:** 6
**Confidence:** 4
**Correctness:** yes
**Clarity:** no

**Review:**

- The proposed method IDSDS is simple but effective, it solves the ood issue when do patch deletion-based evaluation for attribution maps. However, the proposed method can only work on fine-tuned models, which may have very different explainable features compared with the original features.
- The experiments are extensive. 23 attribution methods are tested, showing that intrinsically explainable models are better. e.g. IG better than GradCam.
- The method is too expensive to run, e.g. for ImageNet

**Strengths:**

- Simple methods, easy to follow.
- Shed light on some insights e.g.  larger model is less explainable, removing BN will increase the score
- Did extensive experiments, including 23 tasks.

**Additional Feedback:**

no

**Documentation:**

yes

**Opportunities For Improvement:**

`Review`

**Relation To Prior Work:**

yes

**Summary And Contributions:**

The authors propose a new benchmark to evaluate different attribution methods, they use deletion-based methods, measuring the correspondence between the change in DNN outputs and the masked patches. In order to solve the OOD issue, the authors first train the model on datasets where half of the image samples are masked in patches. The benchmark is model-agnostic and can be used to do inter-model comparison. Testing on extensive models and methods, the authors ranked the attribution methods and provided some insights for the interpretability of DNNs. The paper is overall well-written and easy to understand.

---

> ### Author Rebuttal · Authors · 2024-08-16
>
> We sincerely thank the reviewer for their effort, valuable feedback, and for acknowledging that our paper provides new insights and extensive experiments. We will revise the paper accordingly and address the concerns below.
>
> > "The method is too expensive to run, e.g. for ImageNet"
>
> If we understand this point correctly, there seems to be a misunderstanding. Indeed, our method was applied to the ImageNet dataset (e.g., l.176). We apologize if this has not come across sufficiently clearly, and will revise unclear passages accordingly. If the reviewer meant something else, we would be thankful for a clarification.
>
> > "The proposed method can only work on fine-tuned models, which may have very different explainable features compared with the original features"
>
> We fully agree with this point and recognize it as a valid limitation of our work as discussed in lines 162-166. We agree that the fine-tuned model will learn slightly different features than the original model. However, as we showed in Table 3, our fine-tuning has almost no negative effect on classification accuracy. Thus, a user could also use the fine-tuned model, evaluated with our IDSDS, for their downstream task.
>
> We again thank the reviewer and look forward to further discussion.

---

> > ### Comment · Reviewer_beW6 · 2024-08-16
> > **Thanks for your reply**
> >
> > Q1: I mean we need to fine-tune the model for masked images in order to run the proposed algorithm, will it be expensive for large model / large dataset?
> >
> > Q2: How can we prove the assumption that the model fine-tuned with nearly no performance decrease means they have similar features? A good approach will be visualizing the features / attribution maps.

---

> > > ### Author Rebuttal · Authors · 2024-08-18
> > >
> > > **Q1**: We thank the reviewer for the clarification. In our paper, we train for 30 epochs on the full ImageNet training split (l. 173). With a ResNet-50 and four NVIDIA RTX A6000 (48GB) GPUs (l. 572 in the Appendix), this takes only a couple of hours. For larger datasets and models, the computational cost increases accordingly. However, for larger-scale datasets, we might be able to achieve a domain alignment without using the full dataset. We will include an example of the training time in the revision to give the reader a better estimate of the computational cost. We would also like to note that while our method certainly has a higher run-time than approaches that do not require fine-tuning, we are still orders of magnitude more efficient than ROAR [27], which requires training multiple models per corruption level and attribution method.
> > >
> > > **Q2**: This is a very interesting suggestion that we slightly extend in the following. Since it is hard to show that two models are using similar features, we propose _multiple_ experiments to measure the similarity between two models:
> > >
> > > 1) We measure the mean absolute difference (MAD) between the target softmax outputs of the two models. We suspect that a model using different features will result in different output confidences. Thus, a smaller value indicates more similar models.
> > > 2) Similarly as proposed by the reviewer, we measure the mean absolute difference (MAD) between the GradCAM attributions of the two models. We chose GradCAM for its simplicity and because it is less noisy than many other attribution methods. Intuitively, similar models should yield similar attribution maps, and thus, a smaller value again indicates more similar models.
> > > 3) As also proposed by the reviewer, we randomly select channels from the last convolutional layer of the two models and plot the images that result in the highest activation of that channel. This is an established method to visualize the "concept(s)" learned by a channel and for similar models the shown images should be similar.
> > >
> > > The MADs between the target softmax outputs for the following pairs are:
> > > - Pre-trained ResNet-50 (OOD) and our fine-tuned ResNet-50 (ID): 0.049
> > > - Pre-trained VGG-16 (OOD) and our fine-tuned VGG-16 (ID): 0.028
> > > - Pre-trained ResNet-50 (OOD) and pre-trained VGG-16 (OOD) (this is our baseline): 0.133
> > >
> > > The corresponding OOD and ID models have much more similar target softmax outputs than two different OOD baseline models (ResNet-50 and VGG-16).
> > >
> > > The MADs between the GradCAM attributions for the following pairs are:
> > > - Pre-trained ResNet-50 (OOD) and our fine-tuned ResNet-50 (ID): 0.047
> > > - Pre-trained VGG-16 (OOD) and our fine-tuned VGG-16 (ID): 0.039
> > > - Pre-trained ResNet-50 (OOD) and pre-trained VGG-16 (OOD) (this is our baseline): 0.193
> > >
> > > Again, the corresponding OOD and ID models have much more similar GradCAM attributions than two different OOD baseline models (ResNet-50/VGG-16).
> > >
> > > Finally, we show the highest activating images for three randomly selected channels from the OOD models and our corresponding fine-tuned ID models in the attached PDF. While the order of the images slightly changes, the overall learned concepts seem to be very similar.
> > >
> > > To conclude, in both our quantitative evaluations, the fine-tuned models are significantly more similar to the corresponding OOD model than two different OOD baseline models (ResNet-50/VGG-16). Additionally, the qualitative analysis shows that randomly selected channels react to more or less the same images for the OOD and ID model. These results are a strong indicator that our ID models still use very similar features as the original OOD models, and thus, our proposed method is further validated. We again thank the reviewer for the suggestion and will include the study in the revision.

---

> > > > ### Comment · Reviewer_beW6 · 2024-08-20
> > > > **Thanks for your reply**
> > > >
> > > > Thanks for the response, the additional results to Q2 should be included in the paper, it is very important to make the logic chain reasonable. I will maintain the score towards acceptance and raise my confidence.

---

### Decision · Program_Chairs · 2024-09-26

**Decision:**

Accept (Poster)

**Comment:**

The paper introduces a new benchmark called the In-Domain Single-Deletion Score (IDSDS) to evaluate attribution methods for explaining DNN predictions in computer vision. This benchmark addresses issues such as out-of-domain problems and enables comparisons between different models. The authors propose a model-agnostic approach that ranks attribution methods and provides valuable insights for improving the interpretability of DNNs. Overall, the paper is well-written and contributes to advancing the state-of-the-art in the field of attribution methods in computer vision.